# Somatic Recombination Between an Ancient and a Recent *NOTCH2* Gene Variant Is Associated with the NOTCH2 Gain-of-Function Phenotype in Chronic Lymphocytic Leukemia

**DOI:** 10.3390/ijms252312581

**Published:** 2024-11-22

**Authors:** Rainer Hubmann, Martin Hilgarth, Tamara Löwenstern, Andrea Lienhard, Filip Sima, Manuel Reisinger, Claudia Hobel-Kleisch, Edit Porpaczy, Torsten Haferlach, Gregor Hoermann, Franco Laccone, Christof Jungbauer, Peter Valent, Philipp B. Staber, Medhat Shehata, Ulrich Jäger

**Affiliations:** 1Department of Internal Medicine I, Division of Hematology & Hemostaseology, Medical University of Vienna, 1090 Vienna, Austria; martin.hilgarth@meduniwien.ac.at (M.H.); edit.porpaczy@meduniwien.ac.at (E.P.); peter.valent@meduniwien.ac.at (P.V.); philipp.staber@meduniwien.ac.at (P.B.S.); medhat.shehata@gmail.com (M.S.); ulrich.jaeger@meduniwien.ac.at (U.J.); 2Comprehensive Cancer Center Vienna, Medical University of Vienna, 1090 Vienna, Austria; 3Austrian Red Cross, Blood Service for Vienna, Lower Austria and Burgenland, 1040 Vienna, Austria; andrea.lienhard@roteskreuz.at (A.L.); filip.sima@roteskreuz.at (F.S.); manuel.reisinger@roteskreuz.at (M.R.); claudia.hobel-kleisch@redcross.at (C.H.-K.); christof.jungbauer@redcross.at (C.J.); 4Institute of Medical Genetics, Center for Pathobiochemistry and Genetics, Medical University of Vienna, 1090 Vienna, Austria; tamara.loewenstern@meduniwien.ac.at (T.L.); franco.laccone@meduniwien.ac.at (F.L.); 5MLL Munich Leukemia Laboratory, 81377 Munich, Germany; torsten.haferlach@mll.com (T.H.); gregor.hoermann@mll.com (G.H.); 6Ludwig Boltzmann Institute for Hematology and Oncology, Medical University of Vienna, 1090 Vienna, Austria

**Keywords:** NOTCH2, gain of function, haplotypes, somatic recombination, evolution, CLL

## Abstract

Constitutively active NOTCH2 signaling is a hallmark in chronic lymphocytic leukemia (CLL). The precise underlying defect remains obscure. Here we show that the mRNA sequence coding for the NOTCH2 negative regulatory region (NRR) is consistently deleted in CLL cells. The most common *NOTCH2ΔNRR-DEL2* deletion is associated with two intronic single nucleotide variations (SNVs) which either create (CTT*A*T, G>A for rs2453058) or destroy (CTC*G*T, A>G for rs5025718) a putative splicing branch point sequence (BPS). Phylogenetic analysis demonstrates that rs2453058 is part of an ancient *NOTCH2* gene variant (**1A01*) which is associated with type 2 diabetes mellitus (T2DM) and is two times more frequent in Europeans than in East Asians, resembling the differences in CLL incidence. In contrast, rs5025718 belongs to a recent *NOTCH2* variant (**1a4*) that dominates the world outside Africa. Nanopore sequencing indicates that somatic reciprocal crossing over between rs2453058 (**1A01*) and rs5025718 (**1a4*) leads to recombined *NOTCH2* alleles with altered BPS patterns in *NOTCH2*1A01/*1a4* CLL cases. This would explain the loss of the NRR domain by aberrant pre-mRNA splicing and consequently the NOTCH2 gain-of-function phenotype. Together, our findings suggest that somatic recombination of inherited *NOTCH2* variants might be relevant to CLL etiology and may at least partly explain its geographical clustering.

## 1. Introduction

Chronic lymphocytic leukemia (CLL) is the most common form of adult hematologic malignancy occurring in the Western world, with a five- to ten-fold higher age-adjusted incidence rate in persons of European descent compared to East Asians [1,2,3]. CLL is a blood and bone marrow cancer that typically progresses slowly and primarily affects adults [4]. It is an antigen-driven clonal expansion of neoplastic CD5^+^ B-lymphocytes that consistently overexpress activated NOTCH2 [5,6,7]. The morphogenic stem cell factor NOTCH2 belongs to a family of highly conserved transmembrane receptors that modulate a wide variety of differentiation processes including binary lineage specifications and adult tissue homeostasis [8]. After trans-ligand binding, the NOTCH negative regulatory region (NRR) is stretched, unmasking the S2 cleavage site [9]. This leads to the proteolytic cleavage of S2 by ADAM10 metalloprotease and subsequently of S3 by γ-secretase followed by translocation of the NOTCH intracellular domain (NOTCH^IC^) to the nucleus, where it acts as context-dependent transcription factor on genes like *FCER2* (CD23) in CLL cells [5]. NOTCH2 gain-of-function (GOF) forms, i.e., variants that enhance NOTCH2 activity, are tumorigenic by rendering transformed cells into a less-differentiated immortalized state [10], thereby facilitating the stem cell-like proliferation of cancer cells [11]. Genetic evidence that *Notch2* acts as a CLL-initiating oncogene came from two independent mouse studies. First, Notch2 was found to be indispensable for CLL development in Cd5+ (B-1a) B-cells [12] and, second, enforced expression of truncated, constitutive active Notch2 forms led to the selective development and expansion of Cd5+ (B-1a) B-cells [13]. The genetic basis for constitutively active NOTCH2 signaling in CLL and in many other NOTCH2-associated human malignancies, however, remains to be determined [11].

*NOTCH2* also plays a prominent role in human evolution. A pericentric inversion inv(1)(p12q21.1) involving the *NOTCH2* gene (1p12), together with multiple rounds of gene duplication and gene conversion, distinguishes human chromosome one from its great ape homologs [14,15]. The functional outcome of these genomic rearrangements is human brain size expansion due to enhanced NOTCH2 signaling [15,16]. The possible implication of *NOTCH2* gene variants in CLL leukemogenesis, however, is unknown. Therefore, the aim of this study was to search for *NOTCH2* gene aberrations which might explain the NOTCH2 gain-of-function phenotype in CLL.

Here we show that the constitutive active NOTCH2 phenotype in CLL cells is associated with the expression of aberrant spliced *NOTCH2* mRNAs lacking the sequence coding for the NOTCH negative regulatory region (*NOTCH2ΔNRR*). The most frequent *NOTCHΔNRR* mRNA deletion (DEL2) was found to be linked with somatic reciprocal crossing over between an ancient (**1A01*) and a recent (**1a4*) *NOTCH2* gene variant, leading to recombined *NOTCH2* alleles with mutated splicing branch point sequence (BPS) patterns.

## 2. Results

### 2.1. NOTCH2ΔNRR mRNA Deletions in CLL Cells

One type of mutation which would explain the NOTCH2 GOF phenotype in CLL is deletions affecting the NOTCH2 extracellular domain (NOTCH2^EC^) [17]. Therefore, we screened for *NOTCH2* expression in 20 randomly selected CLL samples (Table 1) at the mRNA level by oligo-dT primed RT-PCR followed by Sanger sequencing. In these experiments, we found recurring *NOTCH2ΔEC* deletions in all CLL cases independent of their clinical stage and prognostic marker profile (Table 1 and Figure 1). Whereas in three patients, the deletions were caused by exon skipping (ES1: E16–27, n = 2; ES2: E24–25, n = 1), the breakpoints from the others (DEL1: E15–26, n = 1; DEL2: E20–27, n = 7; DEL3: E21–26, n = 6; DEL4: E21–28, n = 3; DEL5: E21–26, n = 1; DEL6: E22–26, n = 1; DEL7: E20–26, n = 1; DEL8: E20–26, n = 1) clustered in exonic *NOTCH2* sequences (Table 1 and Figure 1, see Table 2 for relative mRNA, corresponding genomic DNA, and amino acid positions). Their confined localization (5′-hot spots: E15 and E20/21; 3′-hot spots: E26–28) indicates a stringent selection pressure to create a stable NOTCH2 GOF phenotype (Figure 1b). All exonic deletion breakpoints are characterized by the loss of one short direct repeat (DR, Table 1 and Figure 1) indicative of either microhomology-mediated end-joining (MMEJ) after a DNA double strand break [18] or aberrant pre-mRNA splicing [19,20] as the causative genetic event. Interestingly, the majority of *NOTCH2ΔEC* mRNA deletions (ES1, DEL2, DEL3, DEL4, and DEL6) were in frame (17 out of 20, 85%; Table 2), indicating that they were not subjected to degradation by nonsense-mediated mRNA decay in the affected CLL patients. The predicted consequence for the NOTCH2 protein is the loss of the autoinhibitory NOTCH negative regulatory region (NOTCH2ΔNRR) which normally prevents ligand-independent and cis-ligand-mediated NOTCH activation (Figure 1b) [9].

### 2.2. NOTCH2ΔNRR-DEL2 CLL Cases Are Associated with BPS Affecting SNVs

Since corresponding *NOTCH2ΔNRR* deletions could not be detected at the genomic DNA (gDNA) level in CLL cells, we assessed whether aberrant pre-mRNA splicing might be involved in this phenomenon [21,22]. To address this point, we Sanger sequenced the surrounding gDNA of the *NOTCH2ΔNRR* deletion hotspots 5′-HS2 and 3′-HS1 (Figure 1b) with special emphasis on splicing regulatory elements. In 6 out of 7 *NOTCH2∆NRR-DEL2* CLL cases (86%), we found three heterozygous *NOTCH2* SNVs in intron19 (rs2453058), intron21 (rs5025718), and intron26 (rs2793830). The remaining CLL cases were either negative for rs2453058 and rs2793830, or homozygous for rs5025718 (Table 1) indicating that rs2453058 and rs2793830 belong to one and rs5025718 to another *NOTCH2* haplotype. Interestingly, rs2453058 creates (CTT*A*T, G>A), whereas rs5025718 (CTC*G*T, A>G) destroys, a putative splicing branch point sequence (BPS), yUnAy, in the expected genomic context [23].

The minor allele frequency (MAF) of rs2453058, rs2793830, and rs5025718 in European CLL patients (Austria/Spain/Germany, n = 387) is comparable with their average frequency in the normal European population (dbSNP, 1000Genomes_30×) (Figure 2). However, rs2453058 is 2.4 and 6.7 times more frequent in Europeans compared to East Asians (dbSNP, 1000Genomes_30×) and Japanese (jMorp, 54KJPN), respectively, resembling the local differences in CLL incidence (Figure 2) [1,2,3].

### 2.3. The NOTCH2∆NRR-DEL2 Associated SNVs Belong to an Ancient (*1A01) and a Recent (*1a4) NOTCH2 Gene Variant

To estimate the exact structure, evolution, frequencies, and global distribution of the *NOTCH2∆NRR-DEL2*-associated haplotypes [24,25], we analyzed the *NOTCH2* gene in whole genome sequencing (WGS) data from the human genome diversity project (HGDP). This comprises 468 individuals out of 46 ethnicities including the oldest contemporary modern humans, the South African San people [26,27]. The available genomes from three Neanderthals [28,29,30] and one Denisovan [31] from the Max Planck Institute (MPI) for evolutionary anthropology in Leipzig served as the reference for archaic humans (Appendix A). Non-recombining Y chromosome (NRY) haplogroups were used as markers for human demographic history (Appendix A) [32].

Phylogenetic analysis clearly shows that rs245308 belongs to a highly divergent haplotype, which we termed *NOTCH2*1A01* (Figure 3a–d, Appendix A). *NOTCH2*1A01* consists of five closely related subvariants (**1A01*, **1A01a*, **1A01b*, **1A01b1*, and **1A01b2*) that span major parts of *NOTCH2* and the testis-specific expressed *ADAM30* metalloproteinase gene (Figure 3a–c) [33]. Surprisingly, together with the in Oceania prevalent haplotype *NOTCH2*1A02* (Figure 3c), *NOTCH2*1A01* forms an evolutionary old cluster (defined by rs1493696 in intron 11) which precedes the *NOTCH2* gene variants found in Neanderthal (**1A03*) and Denisovan (**1A04*) genomes (Figure 3b and Appendix A) [34,35]. This may suggest that *NOTCH2*1A01* and *NOTCH2*1A02* originate in *Homo erectus* hominins and were, a long time ago, integrated into the modern human gene pool by introgression (dendrogram in Figure 3d) [36,37]. An outstanding feature of *NOTCH2*1A01* is its association with T2DM at nine SNV positions (Figure 3b). The in genome-wide association studies (GWAS) of T2DM frequently identified SNVs rs10923931 (*NOTCH2*) and rs2641348 (*ADAM30*), for example, belong to *NOTCH2*1A01* (Figure 3b).

The evolution of *NOTCH2*1,* which we found at a high frequency in San people and therefore considered it as the first recent *NOTCH2* gene variant (Figure 3c), may be the result of a recombination between the transient *NOTCH2* forms *NOTCH2*1Av2a3* and its progenitor *NOTCH2*1Av2* (Figure 3b–d and Appendix A). *NOTCH2*1* is the last *NOTCH2* variant harboring the *NOTCH2ΔNRR-DEL2*-associated SNV rs2793830 (Haplogroup *1A0-1*, Figure 3b).

Recombination between *NOTCH2*1* and *NOTCH2*1A* may explain the origin of *NOTCH2*1a1*, whose stem form *NOTCH2*1a* evolved further in *NOTCH2*1a2* or *NOTCH2*1a3* (Figure 3b–d and Appendix A). *NOTCH2*1a3* is the African progenitor of *NOTCH2*1a4* which spread all over the world most likely as a result of the latest out-of-Africa bottleneck approximately 60 thousand (K) years ago (Figure 3c,d) [34]. The recent *NOTCH2*1a4* variant dominates Eurasia and America (Figure 3c) and together with *NOTCH2*1a3* includes the third *NOTCH2∆NRR-DEL2*-associated SNV rs5025718 (Figure 3a–d and Appendix A). *NOTCH2*1a4* splits in 16 major subvariants (**1a4v1-16*) with two or three individual markers in most cases (Figure 3b–d and Appendix A).

Consequently, 6 out of 20 CLL cases studied in detail here (30%) possess the ancient/recent *NOTCH2* haplotype combination **1A01/*1a4* which is consistently associated with the recurrent *NOTCH2∆NRR-DEL2* mRNA deletion in CLL cells (Table 1).

### 2.4. NOTCH2 Gene Variant Distribution in CLL Patients

We next studied the *NOTCH2* gene in the available WGS data (International Cancer Genome Consortium, ICGC) from 52 pre-treatment leukemic/non-leukemic CLL samples from a Spanish CLL study cohort (Appendix A) [38,39]. Special attention was paid on the *NOTCH2*1A01/*1a4* haplotype combination and on other CLL-associated *NOTCH2* gene variants in comparison to healthy controls from the HGDP dataset (Figure 3c and Appendix A) [26,27].

Ten Spanish CLL cases (10 out of 52; 19.23%) belong to the *NOTCH2*1A01/*1a4* haplotype group (Appendix A). This corresponds to the percentage of *NOTCH2*1A01/*1a4* cases found in a German CLL validation cohort from the MLL Munich Leukemia Laboratory (57 out of 315; 18.09%). In summary, 18.86% (73 out of 387) of the analyzed European CLL cases (Austria/Spain/Germany) and 16.67% of the European healthy controls (16 out of 96) have the ancient/recent *NOTCH2*1A01/*1a4* haplotype combination (Figure 4). This frequency is two times higher compared to East Asians (7 out of 92; 7.61%), resembling again the local differences in CLL incidence (Appendix A and Figure 4) [1,2,3]. Due to the low frequency of *NOTCH2*1a4* in Africa (Figure 3c and Appendix A), we found no case with the *NOTCH2*1A01/*1a4* combination in analyzed Africans (n = 66, Figure 4).

Two Spanish CLL cases (4%) are heterozygous for the transient/recent *NOTCH2* haplotype combination **1Av2a2/*1a4* (Figure 3b,c and Appendix A), which was otherwise not found in European healthy donors (Appendix A).

Fourteen Spanish CLL cases (27%) are heterozygous (n = 12) or homozygous (n = 2) for the recent *NOTCH2*1a4* subvariant 1 (**1a4v1*), whose two common SNVs are located on the one hand in the corresponding *NOTCH2ΔNRR* minimal deleted region (MDR) in intron 24 (for rs17258579) and on the other hand in the intergenic region between *NOTCH2* and the testis-specific expressed *ADAM30* (for rs115764389) gene (Figure 3b,c and Appendix A). Interestingly, **1a4v1* belongs to those *NOTCH2* gene variants with the pronounced male bias (male/female ratio: 4.5) in CLL (Appendix A).

Nineteen Spanish CLL cases (37%) are heterozygous (n = 18) or homozygous (n = 1) for the recent *NOTCH2*1a4* subvariant 2 (**1a4v2*), whose two common SNVs are located within the corresponding *NOTCH2ΔNRR* deleted region in intron 22 (for rs4659248) and intron23 (for rs72697235), respectively (Figure 3b,c, and Appendix A).

Both *NOTCH2*1a4* subvariants (**1a4v1* and **1a4v2*) peak in their allele frequency in Europe with no major differences to Spanish CLL patients and are almost absent in world regions with low CLL incidence (Figure 3c, and Appendix A) [1,2,3].

All other CLL-associated *NOTCH2* gene variants and SNVs identified in the Spanish WGS dataset together with their frequency (CLL Spain compared to Europe, Africa, East Asia, and Japan), variant allele frequency (VAF), genomic context, gender aspects, and CLL-characteristic *NOTCH1* and *SF3B1* mutations are summarized in Appendix A.

### 2.5. NOTCH2*1A01 and NOTCH2*1a4 Are Recombined in CLL Cells

We hypothesized that the NOTCH2ΔNRR GOF phenotype is induced by somatic homologous recombination [40,41,42] between a maternally and a paternally inherited *NOTCH2* gene variant induced by a DNA double strand (ds) break in a CLL precursor cell.

VAF analysis of individual *NOTCH2* haplotype combinations in the Spanish CLL WGS data (n = 52) revealed a complex picture with many indications of copy-neutral loss of heterozygosity (CN-LOH, VAF ≥ 50 ± 10%) events (Appendix A). However, due to the uncertain significance of the diverse LOH calls [43], we were unable to predict a CLL-specific somatic *NOTCH2* recombination pattern.

Therefore, we concentrated on possible genomic DNA rearrangements in our six Austrian *NOTCH2*1A01/*1a4* CLL cases affecting their allele-specific *NOTCH2* BPS sites (Figure 3b) by haplotype phasing. For this approach, we took advantage of targeted long read sequencing by the Oxford Nanopore technology to capture the relevant SNVs on single reads [44]. This strategy clearly showed that the BPS-creating *NOTCH2*1A01* variant rs2453058 (a) and the BPS-destroying *NOTCH2*1a4* variant rs5025718 (b) are recombined by reciprocal crossing over in all six *NOTCH2*1A01/*1a4* CLL cases leading to mutated rs5025718/rs2453058 (+/+) ba and (−/−) ab alleles (Figure 5a,b). The wild-type variants rs2453058 (a) and rs5025718 (b) were also present in all six CLL cases (Figure 5b), confirming the somatic origin of the recombinations. Three CLL cases were additionally positive for the *NOTCH2*1A01b1* subvariant belonging to rs72697239 (CLL2, 5, 7), which was found in all affected cases on the recombined ab alleles (Figure 5b). Interestingly, the frequency of the four *NOTCH2* alleles is comparable in all six *NOTCH2*1A01/*1a4* CLL cases (Figure 5b), indicative of a balanced distribution within the CLL clones. Moreover, only around 25–50% of CLL cells are heterozygous for the recombined *NOTCH2* variants (Figure 5b).

### 2.6. Detection of NOTCH2*1A01/*1a4 Gene Variants by RFLP

To detect the *NOTCH2*1A01/*1a4* haplotype combination in a routine laboratory setting, we developed a restriction fragment length polymorphism (RFLP) approach. First, we PCR amplified rs2793830 to create an 889 bp fragment where HpyCH4IV single cuts the *NOTCH2*(1a1-)1a4* haplogroup (A*C*GT) at position 396 leaving *NOTCH2*1A0-1* variants uncut. Since the *NOTCH2*1a1-1a3* haplotypes are almost exclusively found in Africa (Figure 3b) we put these variants in brackets. In a second round, we PCR amplified rs2453058 to create a 281bp fragment where PsiI single cuts the *NOTCH2*1A01* allele (TT*A*TAA) at position 95. The expected DNA fragments after a PsiI/HpyCH4IV double-digestion in a case homozygous for *NOTCH2*1a4* compared to the pattern of a cases heterozygous for *NOTCH2*1A01* and *NOTCH2*1a4* (Figure 6a) together with a proof-of-concept RFLP with four healthy donors and 9 CLL samples (Figure 6b) are shown in Figure 6.

## 3. Discussion

The deregulation of NOTCH2 signaling and overexpression of its target gene *FCER2* (CD23) are major characteristics of CLL [1,5,6]. Here we show that this NOTCH2 GOF phenotype is associated with aberrant spliced *NOTCH2* mRNAs lacking the sequence coding for the NOTCH negative regulatory region (*NOTCH2ΔNRR*). The most frequent *NOTCHΔNRR-DEL2* deletion was found to be linked with a recurring reciprocal crossing over between an ancient (**1A01*) and a recent (**1a4*) *NOTCH2* gene variant, leading to recombined *NOTCH2* alleles with mutated BPS patterns.

Somatic homologous recombination between a maternally and paternally inherited chromosome is a rare outcome of the cellular DNA ds-break repair system [40,41]. This led in many cases to CN-LOH by gene conversion, which is associated with most clonally expanded blood cells in the elderly [42]. In the context of our study, however, the *NOTCH2*1a4* and *NOTCH2*1A01* gene variants seemed to be recombined by an even rarer reciprocal crossing-over event, where both mutated *NOTCH2* alleles are maintained without significant changes in the VAF values of the affected SNVs. The distribution of the wild-type and mutated *NOTCH2* alleles within the CLL clone demonstrate not only the somatic origin and balanced nature of this recombination but indicates also that only around 20–50% of the leukemic cells are heterozygous for the recombined *NOTCH2* alleles. This might be explained by the instructive properties of NOTCH signaling (lateral induction), a mechanism aimed to synchronize the fate of a particular cell population [8]. Furthermore, the crossing over within the *NOTCH2* gene together with the described male bias for intragenic recombination [45] would be a plausible reason why men are more often affected by CLL than women are.

Somatic recombination might be also relevant for other CLL-associated *NOTCH2ΔNRR* mRNA deletions and *NOTCH2* gene variant combinations found in this work and will therefore be analyzed in prospective haplotype phasing studies with an expanded study cohort. The in Europe prevalent *NOTCH2*1a4* subvariants **1a4v1* and **1a4v2*, for example, account for 62% of Spanish CLL cases analyzed. Together with the distribution of the *NOTCH2*1A01/*1a4* combination studied in detail in this work (Figure 4), the local frequencies of *NOTCH2*1a4v1* and **1a4v2* (Figure 3c,d) resemble the differences in CLL incidence between persons of European descent compared to East Asians [1,2,3], making these recent *NOTCH2*1a4* subvariants additional candidates for hereditary and geographical aspects of CLL. Their relative high frequency in the normal European population, however, may limit their clinical relevance as CLL predisposition markers, if local differences are not taken into consideration. This may explain otherwise why *NOTCH2*-associated SNVs have not been considered in CLL genome-wide association studies (GWAS) [46].

The origin of ancient *NOTCH2* gene variants in contemporary humans remains elusive and their assignment to certain archaic hominins at this point is purely speculative. However, our phylogenetic analysis clearly shows that *NOTCH2*1A01* and *NOTCH2*1A02* belong to haplotypes older then the archaic *NOTCH2* variants found in Neanderthals (**1A03*) and Denisovans (**1A04*). Considering their global distribution and high number of individual SNVs reflecting their relative age/distance to their stem form **1A0* (Figure 3c and Appendix A), we hypothesize that these ancient *NOTCH2* gene variants may trace back to African *Homo erectus* hominins, were early integrated into the modern human gene pool by introgression [34,35,36,37], and maintained by local selection advantages and/or balanced polymorphisms [47]. Their assumed African origin is also supported by the presence of pre-variants in certain African ethnicities *(*1A01pre* in Biaka and **1A02pre* in Bantu people, Figure 3c).

A common feature of all *NOTCH2* variants identified so far is their unexpected global conservation, with only few indications of meiotic recombination. This restricted recombination rate is a typical characteristic of the evolutionary dead-end properties of “supergenes”, which are tightly linked loci that control complex balanced polymorphisms within populations [48,49]. The genomic linkage of *NOTCH2* with the testis-specific expressed metalloproteinase *ADAM30*, for example, indicates a yet-to-be-defined function in spermatogenesis. The peak of the ancient *NOTCH2* variants **1A01* (Mandenka, Yoruba, Bantu, Biaka, Mbuti) and **1A02* (Bougainville, Papuan, Karitiana) in equator regions (Figure 3c) suggests a selection advantage in the context of intense sun exposure. Finally, the striking association of *NOTCH2*1A01* with T2DM at nine SNV positions (Figure 3b) might be simply a consequence of the dietary habits of its archaic human source population. The few examples of local meiotic recombination between *NOTCH2* gene variants identified in our analysis (Figure 3d) presuppose a compatible genetic background and, thus, may reflect those geographic regions where the original hominin groups met and interbred [34,35,36,37]. However, this complex topic is beyond the scope of this work and should be analyzed in further studies.

Considering that *NOTCH2* is a tumor-inducing oncogene [10] and indispensable for CLL development in a mouse model [12], we hypothesize that the induction of a NOTCH2ΔNRR GOF phenotype by somatic recombination has the potential to be an initial malignant hit in a CLL precursor cell. Therefore, this work gains fundamental insights into CLL leukemogenesis, including hereditary (recombination between inherited *NOTCH2* gene variants), gender (male bias in CLL patients), and geographical aspects (local differences in CLL incidence), and sheds light on an unexpected role of *NOTCH2* in human evolution and global dispersal. The results might also pave the way for the development of diagnostic and prognostic tools aimed to identify persons with a predisposition for CLL and other NOTCH2-associated tumors as part of a personalized medicine concept and to track the progression of CLL in patients with specific *NOTCH2* gene variants in longitudinal studies.

## 4. Materials and Methods

### 4.1. Patients’ Characteristics and Sample Collection

Peripheral blood mononuclear cells (PBMC) were isolated using Ficoll-Hypaque (GE Healthcare, Uppsala, Sweden) centrifugation. Twenty randomly chosen CLL cases (n = 20) were consecutively sampled and screened for CLL-characteristic chromosomal aberrations by FISH analysis. The exclusion criterion was samples with less than 95% leukemic cells (CD5+/CD19+). The *IGHV* and *NOTCH1* mutational status was determined by Sanger sequencing. The patients’ characteristics are shown in Table 1.

### 4.2. gDNA and RNA Extraction, (RT-) PCR, and Sanger Sequencing

Total RNA was extracted using the TRI Reagent^®^ isolation system (Sigma-Aldrich, St Louis, MO, USA). Total gDNA was isolated using and the Wizard^®^ genomic DNA purification kit (Promega, Madison, WI, USA). Oligo-dT primed RNA was reverse transcribed into cDNA by Moloney murine leukemia virus (M-MLV) reverse transcriptase. GoTaq green master mixes were used for PCR (Promega, Madison, WI, USA). The NCBI Primer-BLAST program (https://www.ncbi.nlm.nih.gov/tools/primer-blast/ accessed on 19 November 2024) based on Primer3 (https://github.com/primer3-org/primer3 accessed on 19 November 2024, version 2.5.0) was applied for Primer design (accessed on 12 March 2021). The following primers were used for RT-PCR analysis of *NOTCH2ΔNRR* mRNA: N2EClong (for DEL1 and ES1) forward 5′-CCTTTCGAATCCATGCCAGA-3′ and reverse 5′-GACGCTTGTGATTGCTTGCA-3′; and N2ECshort (for DEL2-8 and ES2) forward 5′-TGGGCTGGTGCCTATTGTGA-3′ and reverse 5′-CTGGGGCCCTTCATCATCGA-3′. PCR bands were stained with GelRedTM (Biotium, Fremont, CA, USA) and visualized using the ChemiDoc^TM^ gel imaging system from Bio-Rad (Hercules, CA, USA). PCR products were isolated using the illustra^TM^ GFX^TM^ Gel Band Purification Kit (Avantor, Wayne, PA, USA). Sanger sequencing was performed on a 3730 XL DNA analyzer from Applied Biosystems (LGC Genomics, UK). Sequence analysis was conducted taking advantage of the nucleotide blast program (https://blast.ncbi.nlm.nih.gov/Blast.cgi accessed on 19 November 2024).

### 4.3. Genomic DNA Sequence Analysis and WGS Datasets

Aligned WGS reads from contemporary (468 individuals out of 46 ethnicities) [26,27] and archaic (three Neanderthals and one Denisovan) [28,29,30,31] humans were obtained from the human genome diversity project (HGDP) [26,27] deposited by the International Genome Sample Resource (IGSR; https://www.internationalgenome.org/ 19 November 2024) and from the Max Planck Institute for evolutionary anthropology (https://www.eva.mpg.de/genetics/genome-projects/ accessed on 19 November 2024). Aligned WGS reads from 52 Spanish CLL patients [38,39] were obtained from the International Cancer Genome Consortium (ICGC; http://platform.icgc-argo.org accessed on 19 November 2024) [50]. The WGS dataset from 315 German CLL patients from the MLL Munich Leukemia Laboratory provided validation for the European *NOTCH2*1A01/*1a4* combination frequency analysis (Torsten Haferlach, personal communication). *NOTCH2* gene variants and their frequency in various populations were manually determined by identifying alleles with equal SNV patterns (n ≥ 3) using the Integrative Genomics Viewer (http://software.broadinstitute.org/software/igv/ accessed on 19 November 2024) [51] version 2.1.2 together with the NCBI dbSNP Database (1000Genomes_30×, https://www.ncbi.nlm.nih.gov/snp/ accessed on 19 November 2024) and the Japanese Multi Omics Reference Panel (jMorp, 54KJPN, https://jmorp.megabank.tohoku.ac.jp/ accessed on 19 November 2024). The distribution of *NOTCH2* gene variants in human ethnicities and in CLL patients was manually analyzed according to their minor allele frequency (MAF) values.

### 4.4. Haplotype Phasing

Haplotype phasing was conducted by targeted single-molecule real-time sequencing using the Oxford Nanopore sequencing technology [44]. PCR primers amplifying the genomic *NOTCH2* DNA region spanning rs5025718 (**1a4*), rs72697239 (*NOTCH2*1A01b1*), and rs2453058 (**1A01*) read as follows: Primary PCR: N2-1A01-1a4-P forward 5′-GCTTAACCTGTTGGGTTTCCCT-3′ and reverse 5′-TCAGAGTAGGCCCCAGCATA-3′ (4567 bp); nested PCR: N2-1A01-1a4-N forward 5′-AACAATGTGGTACAAAACCACCC-3′ and reverse 5′-ATTTCATACCCCGAGTGCCT-3′ (3152 bp). Samples were prepared for nanopore sequencing with the Native barcoding Kit (SQK-NBD114.24) and sequenced for 20 h on a GridION device (Oxford Nanopore Technologies, Avantor, San Francisco, CA, USA). Binary Alignment Map (BAM)-files (human genome assembly GRCh38) were generated using Samtools (http://www.htslib.org/ accessed on 19 November 2024) version 1.21 and phased with the Integrative Genomics Viewer (IGV). The distribution of *NOTCH2* gene variants within individual CLL samples was manually analyzed according to their variant allele frequency (VAF) values.

### 4.5. RFLP

The NCBI Primer-BLAST software (https://www.ncbi.nlm.nih.gov/tools/primer-blast/ accessed on 19 November 2024) based on Primer3 (https://github.com/primer3-org/primer3 accessed on 19 November 2024, version 2.5.0) together with the NEBcutter tool (http://nc2.neb.com/NEBcutter2/ accessed on 19 November 2024) version V2.0 were used for RFLP design aimed to create PCR amplicons, where rs2453058 and rs2793830 are specifically cut with the restriction enzymes HpyCH4IV (for rs2793830) and PsiI (for rs24530058). The NOTCH2-RFLP primers read as follows: rs2453058 forward 5′-GGTGTACCTCGGAACACTGA-3′ and reverse 5′-ACAATAGGCACCAGCCCATC-3′; rs2793830 forward 5′-CCTGAGAACCTGGCAGAAGG-3′ and reverse 5′-CCCACCTTCCTTAGTCTGGC-3′. The restriction enzymes were purchased from New England BioLabs (Frankfurt am Main, Germany). 60 ng gDNA was used for PCR reactions with the following conditions: 95 °C, 3′; (95 °C, 20″; 56 °C, 20″; 72 °C, 1′) × 38; 72 °C, 10′.

## Figures and Tables

**Figure 1 ijms-25-12581-f001:**
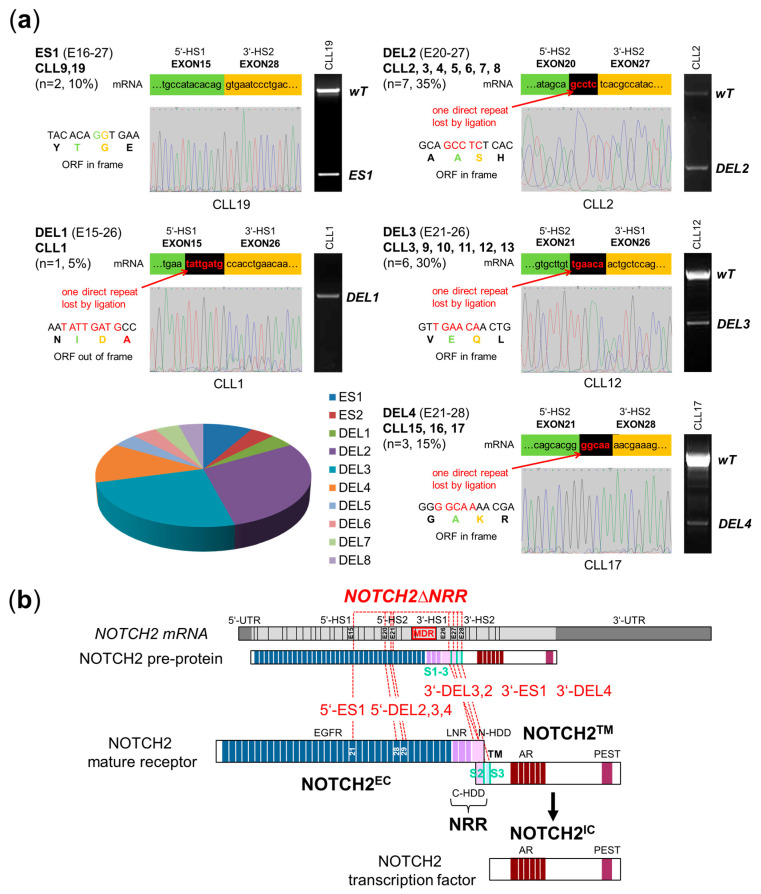
*NOTCH2ΔNRR* mRNA deletions in CLL. (**a**) Representative RT-PCR and sanger sequencing of the most frequent *NOTCH2ΔNRR* mRNA deletions in CLL cells. The fused *NOTCH2* mRNA/protein sequences, the short direct repeat at the deletion breakpoints, and a pie chart showing the frequency of individual *NOTCH2∆NRR* mRNA deletions in 20 CLL cases are indicated. (**b**) Localization of *NOTCH2ΔNRR* deletions at the *NOTCH2* mRNA and corresponding protein level. The S1–3 cleavage sites, the deletion hotspots (HS), and the minimal deleted region (MDR: E24 and 25) are indicated. Abbreviations: UTR, untranslated region; EGFR, epidermal growth factor-like repeats; LNR, Lyn/Notch repeats; HDD, hetero dimerization domain; TM, transmembrane domain; AR, ankyrin repeats; PEST, pest domain.

**Figure 2 ijms-25-12581-f002:**
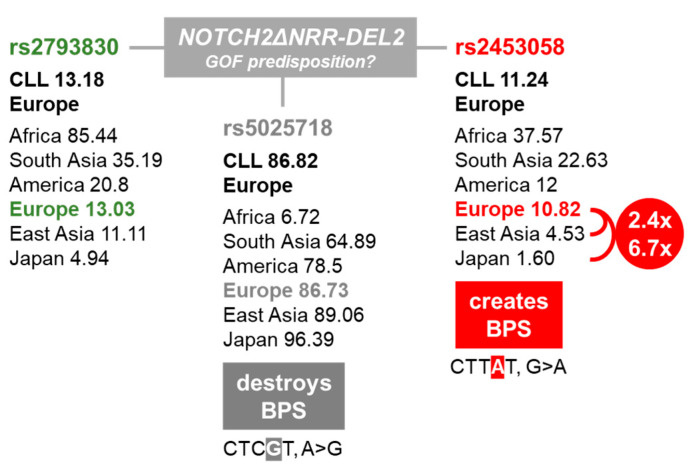
Minor allele frequency (MAF %) of *NOTCH2ΔNRR-DEL2*-associated SNVs in European CLL patients (Austria/Spain/Germany, n = 387) compared to different world regions (1000Genomes_30×; 54KJPN).

**Figure 3 ijms-25-12581-f003:**
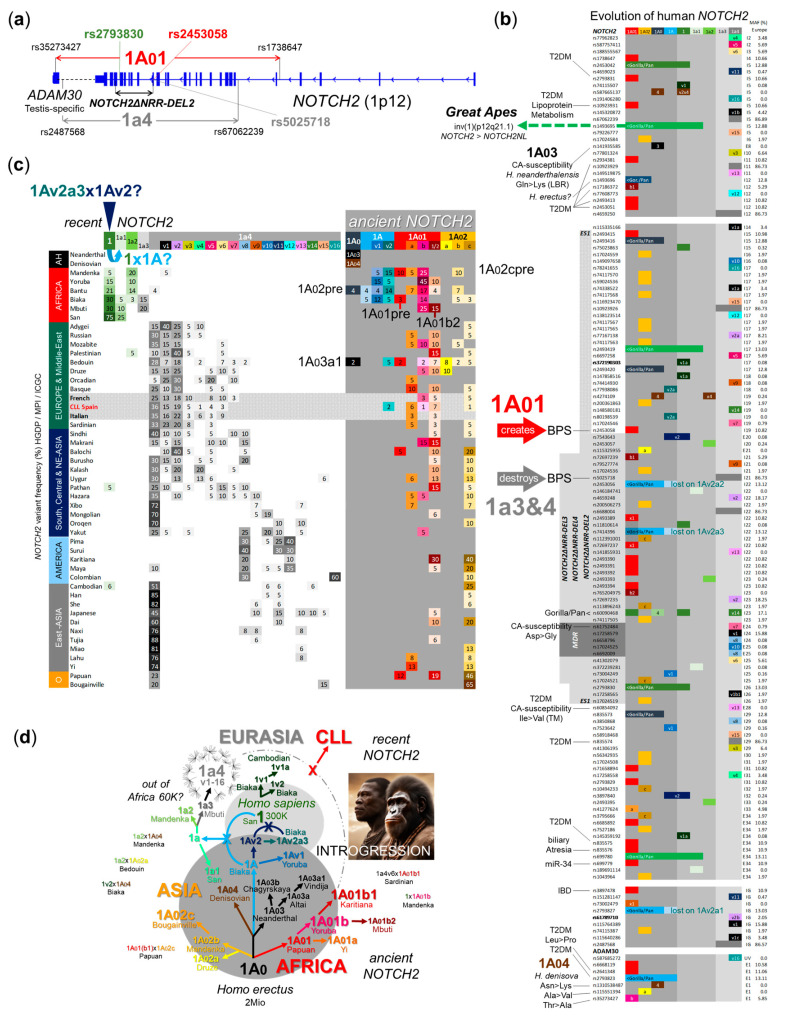
Evolution of the human *NOTCH2* gene. (**a**) Genomic context of *NOTCH2ΔNRR-DEL2*-associated SNVs in European CLL patients (Austria/Spain/Germany, n = 387) compared to different world regions (1000Genomes_30×; 54KJPN). (**b**) *NOTCH2* variants and haplogroups (IG, intergenic), MAF % in Europe, and associated diseases (dbSNP). SNVs shared with great apes are indicated (IBD, Inflammatory bowel disease; LBR, ligand-binding region). (**c**) Global distribution of *NOTCH2* variants in 46 contemporary ethnicities (HGDP; O, Oceania) and archaic humans (AH) compared to Spanish CLL patients (ICGC). Ethnicities with *NOTCH2* pre-variants are indicated. (**d**) Ethnicities with their highest frequency, putative local recombination events, and the 60 K out of Africa bottleneck are indicated. (**c**,**d**).

**Figure 4 ijms-25-12581-f004:**
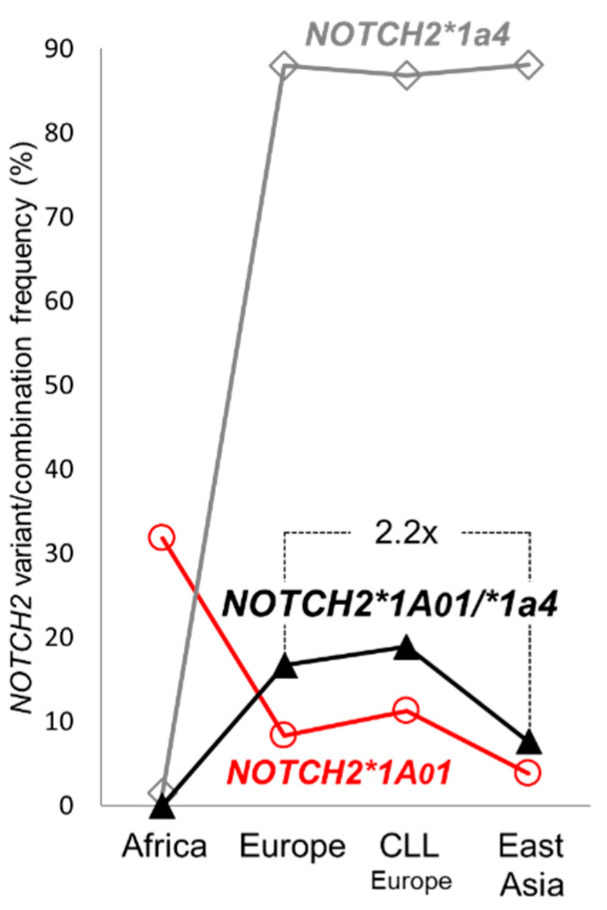
Global distribution of the CLL relevant *NOTCH2*1A01/*1a4* haplotype combination. Frequencies of *NOTCH2*1A01* and **1a4* haplotypes (including all subvariants), and their combination in Africans (HGDP, n = 66), Europeans (HGDP, n = 96), European CLL patients (Austria/Spain/Germany, n = 387), and East Asians (HGDP, n = 92). The 2.2-fold higher frequency of *NOTCH2*1A01/*1a4* in Europeans compared to East Asians is indicated.

**Figure 5 ijms-25-12581-f005:**
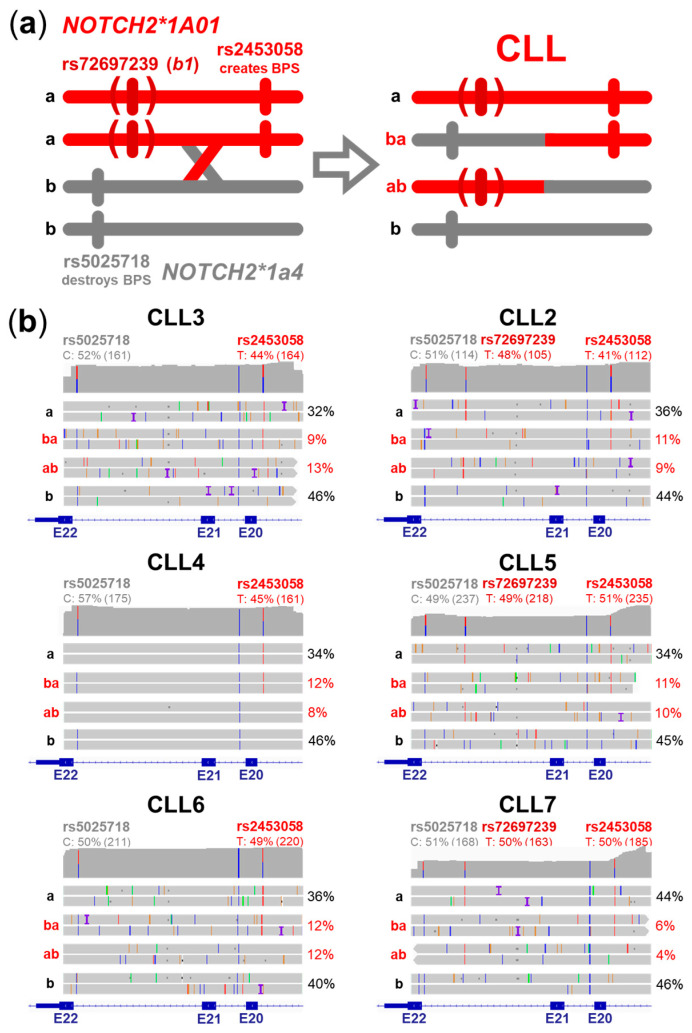
Somatic recombination of *NOTCH2* in CLL. (**a**) Reciprocal crossing over between (a) *NOTCH2*1A01(b1)* and (b) *NOTCH2*1a4* led to recombined *NOTCH2* alleles (ba and ab) with mutated BPS pattern in CLL cells. (**b**) Nanopore single-read alignments (IGV) of six *NOTCH2*1A01(b1)/*1a4* CLL cases showing two representative reads of wild-type (a and b) and recombined (ba and ab) alleles in each case. VAFs of the affected SNVs, read depth (in brackets), the percentage of wild-type and recombined *NOTCH2* alleles, and exons are indicated.

**Figure 6 ijms-25-12581-f006:**
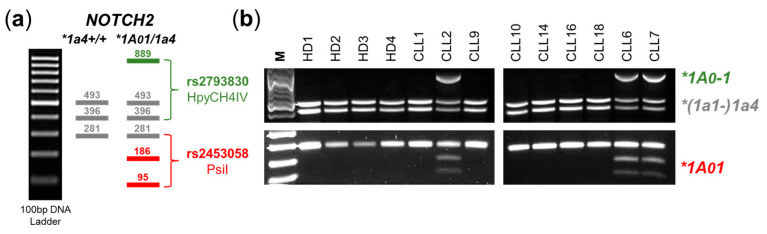
*NOTCH2*1A01/*1a4*-RFLP. (**a**) Expected DNA fragments of HpyCH4IV/PsiI digested rs2453058/rs2793830-PCR products in cases homozygous for *NOTCH2*1a4* and heterozygous for *NOTCH2*1A01/1a4*. (**b**) Proof of concept RFLP showing the expected DNA fragments in 4 HD and 9 CLL samples.

**Table 1 ijms-25-12581-t001:** Clinical and prognostic parameters of CLL patients in relation to *NOTCH2ΔNRR* mRNA deletions.

ID	Age	Gender	R/B	IGHV Mut. (Clone)	Cytogenetic	*NOTCH1* Mutations	*NOTCH2* mRNA Deletions	Direct Repeat *	SNV rs2453058 Intron19 ^†^	SNV rs5025718 Intron21 ^†^	SNV rs2793830 Intron26 ^†^
CLL1	75	m	I/A	U (VH1-69)	13q- (8%), 11q- (14%)	wt	DEL1	TATTGATG	−/−	+/+	−/−
CLL2	68	f	II/B	U (VH1-2)	13q- (80%)	NOTCH1ΔC	DEL2	GCCTC	+/−	+/−	+/−
CLL3	65	m	I/A	U (VH1-69)	normal	NOTCH1ΔC	DEL2 DEL3	GCCTC TGAACA	+/−	+/−	+/−
CLL4	58	m	II/B	M (VH3-43)	normal	wt	DEL2	GCCTC	+/−	+/−	+/−
CLL5	55	m	IV/C	M (VH3-21)	13q- (35%), 11q- (91%)	wt	DEL2	GCCTC	+/−	+/−	+/−
CLL6	67	m	IV/C	U (VH1-69)	13q- (20%)	wt	DEL2	GCCTC	+/−	+/−	+/−
CLL7	81	m	II/B	U (VH1-3)	Tri12 (80%)	wt	DEL2	GCCTC	+/−	+/−	+/−
CLL8	75	m	II/B	U (VH2-5) U (VH3-11)	13q- (8%)	NOTCH1ΔC	DEL2 DEL8	GCCTC TG	−/−	+/+	−/−
CLL9	80	m	I/A	U (VH1-2) U (VH3-9)	normal	wt	DEL3 ES1	TGAACA	−/−	+/+	−/−
CLL10	64	m	I/A	U (VH5)	13q-(13%)	wt	DEL3	TGAACA	−/−	+/+	−/−
CLL11	39	f	I/A	U (VH3-20)	normal	wt	DEL3	TGAACA	−/−	+/+	
CLL12	82	m	I/A	U (VH5-51)	normal	wt	DEL3	TGAACA	−/−	+/+	−/−
CLL13	67	m	I/A	M (VH3-48)	normal	wt	DEL3	TGAACA	−/−	+/+	
CLL14	50	f	IV/C	U (VH1-69)	normal	wt	DEL5	TGCCA	−/−	+/+	−/−
CLL15	63	m	IV/C	M (VH3-23)	13q- (75%)	wt	DEL4	GGCAA	−/−	+/+	−/−
CLL16	60	m	II/B	U (VH1-69)	13q- (13%)	wt	DEL4	GGCAA	−/−	+/+	−/−
CLL17	60	m	IV/C	U (VH6-1)	13q- (10%), Tri12 (10%)	NOTCH1ΔC	DEL4	GGCAA	−/−	+/+	−/−
CLL18	51	m	II/B	U (VH3-11)	13q- (70%), 11q- (70%)	wt	DEL6 DEL7	ACTGTG CAG/ACCT	−/−	+/+	−/−
CLL19	65	m	IV/C	M (VH1-8)	normal	ND	ES1		−/−	+/+	−/−
CLL20	61	f	I/A	U (VH2-5)	normal	ND	ES2		−/−	+/+	−/−

The affected * direct repeats flanking the individual *NOTCH2ΔNRR* deletions and the *NOTCH2ΔNRR-DEL2*-associated ^†^ SNVs are indicated. Abbreviations: R/B, Rai/Binet stage; U/M, *IGHV* unmutated/mutated; ES, exon skipping.

**Table 2 ijms-25-12581-t002:** Relative genomic DNA (NG_008163.2), mRNA (NM_024408.4), and amino acid (NP_077719.2) positions of individual *NOTCH2ΔNRR* mRNA deletions.

*NOTCH2* mRNA Deletions	5′-*NOTCH2ΔNRR* Breakpoint mRNA/gDNA (Exon) *	3′-*NOTCH2ΔNRR* Breakpoint mRNA/gDNA (Exon) *	ORF after Fusion (aa Position) ^†^
DEL1	2648/123896(E15)	4915/150809 (E26)	frameshift (797–Ter)
DEL2	3585/136785 (E20)	5210/151967 (E27)	in frame (11097–1652)
DEL3	3609/137197 (E21)	4926/150823 (E26)	in frame (11177–1557)
DEL4	3743/137333 (E21)	5357/152303 (E28)	in frame (11627–1701)
DEL5	3618/137205 (E21)	4918/150811 (E26)	frameshift (11207–Ter)
DEL6	3808/130075 (E22)	4859/150756 (E26)	in frame (11847–1535)
DEL7	3584/136783 (E20)	4986/150883 (E26)	frameshift (11087–Ter)
ES1 (E16–27)	2734/123930 (E15)	5257/152208 (E28)	in frame (8267–1669)
ES2 (E24&25)	4148/145678 (E23)	4767/150670 (E26)	frameshift (12977–Ter)

* The affected exons and the predicted consequence for the open reading frame ^†^ (ORF) after fusion is indicated.

## Data Availability

The Oxford Nanopore sequencing datasets generated during the current study have been deposited at the NCBI Sequence Read Archive (SRA) under accession number PRJNA1095708 (https://www.ncbi.nlm.nih.gov/sra/PRJNA1095708 accessed on 19 November 2024).

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
