# Peer review of "Somatic Recombination Between an Ancient and a Recent *NOTCH2* Gene Variant Is Associated with the NOTCH2 Gain-of-Function Phenotype in Chronic Lymphocytic Leukemia"

_ijms, 2024, doi:10.3390/ijms252312581_

Round 1
Reviewer 1 Report
Comments and Suggestions for Authors
Dear Editor,
After a thorough review of the manuscript titled "Somatic recombination between an ancient and a recent NOTCH2 gene variant is associated with the NOTCH2 gain-of-function phenotype in Chronic Lymphocytic Leukaemia," I would like to commend the authors for their well-structured and highly relevant study. This research addresses an intriguing aspect of chronic lymphocytic leukemia (CLL) by exploring the somatic recombination of NOTCH2 gene variants and its connection to the gain-of-function phenotype in CLL, particularly within a genetic and geographical context.
Strengths of the Study
Relevance and Originality: The study investigates a novel link between somatic recombination and CLL, specifically associating NOTCH2 variants with the disease phenotype. This innovative approach offers a fresh perspective on the etiology of CLL.
Robust Methodology: The combination of RT-PCR, Sanger sequencing, Nanopore long-read sequencing, and RFLP effectively supports the study's conclusions. These methods are appropriate for detecting genetic recombination and are well-described in several sections of the manuscript.
Suggestions for Improvement
Expansion of Structural Details in the Introduction: It is recommended to expand the section discussing structural genomic alterations (gene duplications, conversions) in the human NOTCH2 gene. Providing more context about the evolutionary significance of the ancient (1A01) and recent (1a4) NOTCH2 gene variants would enhance accessibility for readers less familiar with these concepts.
Clarification on Patient Recruitment and Selection: In the Methods section, it would be beneficial to describe the patient recruitment process in greater detail, specifying the inclusion and exclusion criteria, the sampling method (e.g., consecutive sampling), and how the 20 CLL cases were selected for the study. This would provide better transparency regarding the representativeness of the sample. Although the authors mention in the Results section that "we screened for NOTCH2 expression in 20 randomly selected CLL samples," this information should also be included in the Methods section to ensure clarity and completeness in the description of sample selection.
Enhancement of Table 1: The current format of Table 1 appears to be directly copied from an Excel spreadsheet and could benefit from a more polished, professional presentation to align with the journal's standards. Additionally, some abbreviations, such as "R/B," are undefined and should be clarified for the reader's ease of interpretation. Splitting the table into two, if necessary, and ensuring that all abbreviations are defined would enhance its quality and accessibility.
Simplification and Clarification of Figure 2: Figure 2 contains a considerable amount of information, which can make it challenging for readers to interpret effectively. It is suggested to divide Figure 2 into three separate figures or simplify it by moving some of the detailed data into the main text. Consider dividing the figure according to the specific subsections of the Results section, aligning with the topics:
"NOTCH2ΔNRR-DEL2 CLL cases are associated with BPS affecting SNV"
"The NOTCH2ΔNRR-DEL2 associated SNVs belong to an ancient (1A01) and a recent (1a4) NOTCH2 gene variant"
"NOTCH2 gene variant distribution in CLL patients"
Enhancing the figure resolution would also improve readability. This approach would streamline the figures and enhance clarity, making it easier for readers to focus on the most critical aspects of the data presented.
Specification of Analyzed Variables in the Methods Section: The Methods section associated with "NOTCH2 gene variant distribution in CLL patients" describes how and from where the data were obtained but does not specify all the variables that were analyzed (e.g., variant allele frequencies (VAF), haplotype combinations and specific gene variants, single nucleotide variants (SNVs) associated with NOTCH2) and why they were analyzed. While some descriptions are provided in the Results section, it is essential that the variables analyzed are clearly stated in the Methods section, along with how and why they were analyzed. Ensuring that all variables discussed in the Results have corresponding statements in the Methods about their analysis procedures would enhance the clarity and reproducibility of the study.
Detailed Description of the RFLP Method: The description of the RFLP method lacks crucial details necessary for experimental replication. Specifically, the thermocycling conditions, equipment used, reagent concentrations, amounts of DNA, reaction volumes, and other procedural details are not provided. Including a complete PCR protocol with all relevant parameters in the Methods section is essential to allow other researchers to reproduce the experiments accurately.
Overall Organization and Flow of the Study: The general organization of the study is somewhat confusing, with mixing of topics and abrupt changes that can make it difficult for readers to follow the progression of the research. It is recommended to consider reorganizing the study to introduce topics more smoothly, providing explanations for why certain analyses were conducted, and avoiding sudden shifts between topics. This would improve the readability of the manuscript and help guide the reader through the logical flow of the study.
Focus the Discussion on Main Findings: The current discussion intermixes evolutionary aspects with the study's results, which may confuse readers. It is recommended to focus the discussion on the main findings of the study—specifically, that the NOTCH2 gain-of-function (GOF) phenotype is associated with aberrantly spliced NOTCH2 mRNAs lacking the sequence coding for the NOTCH negative regulatory region (NOTCH2ΔNRR). Emphasizing how the most frequent NOTCHΔNRR-DEL2 deletion is linked to a recurring reciprocal crossing over between ancient (1A01) and recent (1a4) NOTCH2 gene variants would provide clearer context and better prepare readers for understanding the results.
Conclusion
This study represents a significant contribution to the field of molecular genetics and CLL research, offering new insights into how somatic recombination can influence disease predisposition and presentation. By addressing the suggestions outlined above, the manuscript would be strengthened considerably, enhancing its clarity, coherence, and impact on the scientific community. These improvements could have substantial implications for understanding genetic risk factors in CLL and potentially informing early detection strategies for high-risk populations.
Thank you for the opportunity to review this manuscript. I look forward to its contributions to the literature on CLL and genetic predispositions.
Sincerely,
Author Response
Dear Editor,
After a thorough review of the manuscript titled "Somatic recombination between an ancient and a recent NOTCH2 gene variant is associated with the NOTCH2 gain-of-function phenotype in Chronic Lymphocytic Leukaemia," I would like to commend the authors for their well-structured and highly relevant study. This research addresses an intriguing aspect of chronic lymphocytic leukemia (CLL) by exploring the somatic recombination of NOTCH2 gene variants and its connection to the gain-of-function phenotype in CLL, particularly within a genetic and geographical context.
Strengths of the Study
Relevance and Originality: The study investigates a novel link between somatic recombination and CLL, specifically associating NOTCH2 variants with the disease phenotype. This innovative approach offers a fresh perspective on the etiology of CLL.
Robust Methodology: The combination of RT-PCR, Sanger sequencing, Nanopore long-read sequencing, and RFLP effectively supports the study's conclusions. These methods are appropriate for detecting genetic recombination and are well-described in several sections of the manuscript.
Suggestions for Improvement
Expansion of Structural Details in the Introduction: It is recommended to expand the section discussing structural genomic alterations (gene duplications, conversions) in the human NOTCH2 gene. Providing more context about the evolutionary significance of the ancient (1A01) and recent (1a4) NOTCH2 gene variants would enhance accessibility for readers less familiar with these concepts.
Response: We expanded the introduction with the sentence "The possible implication of NOTCH2 gene variants in CLL leukemogenesis, however, is unknown."
Clarification on Patient Recruitment and Selection: In the Methods section, it would be beneficial to describe the patient recruitment process in greater detail, specifying the inclusion and exclusion criteria, the sampling method (e.g., consecutive sampling), and how the 20 CLL cases were selected for the study. This would provide better transparency regarding the representativeness of the sample. Although the authors mention in the Results section that "we screened for NOTCH2 expression in 20 randomly selected CLL samples," this information should also be included in the Methods section to ensure clarity and completeness in the description of sample selection.
Response: The patient recruitment strategy is now described in detail as suggested.
Enhancement of Table 1: The current format of Table 1 appears to be directly copied from an Excel spreadsheet and could benefit from a more polished, professional presentation to align with the journal's standards. Additionally, some abbreviations, such as "R/B," are undefined and should be clarified for the reader's ease of interpretation. Splitting the table into two, if necessary, and ensuring that all abbreviations are defined would enhance its quality and accessibility.
Response: We devided Table 1 in two parts and transferred the sequence details in the method section (Table 2). The format is know according the journal guidelines. R/B is now explained.
Simplification and Clarification of Figure 2: Figure 2 contains a considerable amount of information, which can make it challenging for readers to interpret effectively. It is suggested to divide Figure 2 into three separate figures or simplify it by moving some of the detailed data into the main text. Consider dividing the figure according to the specific subsections of the Results section, aligning with the topics:
"NOTCH2ΔNRR-DEL2 CLL cases are associated with BPS affecting SNV"
"The NOTCH2ΔNRR-DEL2 associated SNVs belong to an ancient (1A01) and a recent (1a4) NOTCH2 gene variant"
"NOTCH2 gene variant distribution in CLL patients"
Enhancing the figure resolution would also improve readability. This approach would streamline the figures and enhance clarity, making it easier for readers to focus on the most critical aspects of the data presented.
Response: Figure 2 is now subdivided in Figure 2 and Figure 3, and simplified, as recommended. A higher resolution of the figures is provided to the journal.
Specification of Analyzed Variables in the Methods Section: The Methods section associated with "NOTCH2 gene variant distribution in CLL patients" describes how and from where the data were obtained but does not specify all the variables that were analyzed (e.g., variant allele frequencies (VAF), haplotype combinations and specific gene variants, single nucleotide variants (SNVs) associated with NOTCH2) and why they were analyzed. While some descriptions are provided in the Results section, it is essential that the variables analyzed are clearly stated in the Methods section, along with how and why they were analyzed. Ensuring that all variables discussed in the Results have corresponding statements in the Methods about their analysis procedures would enhance the clarity and reproducibility of the study.
Response: A detailed description of the analyzed variables and how the analyzis were conducted is now provided in the method section.
Detailed Description of the RFLP Method: The description of the RFLP method lacks crucial details necessary for experimental replication. Specifically, the thermocycling conditions, equipment used, reagent concentrations, amounts of DNA, reaction volumes, and other procedural details are not provided. Including a complete PCR protocol with all relevant parameters in the Methods section is essential to allow other researchers to reproduce the experiments accurately.
Response: A detailed protocoll for the RFLP assay can now be found in the method section.
Overall Organization and Flow of the Study: The general organization of the study is somewhat confusing, with mixing of topics and abrupt changes that can make it difficult for readers to follow the progression of the research. It is recommended to consider reorganizing the study to introduce topics more smoothly, providing explanations for why certain analyses were conducted, and avoiding sudden shifts between topics. This would improve the readability of the manuscript and help guide the reader through the logical flow of the study.
Response: We tried our best to introduce topics more smoothly to improve the readability of our manuscript.
Focus the Discussion on Main Findings: The current discussion intermixes evolutionary aspects with the study's results, which may confuse readers. It is recommended to focus the discussion on the main findings of the study—specifically, that the NOTCH2 gain-of-function (GOF) phenotype is associated with aberrantly spliced NOTCH2 mRNAs lacking the sequence coding for the NOTCH negative regulatory region (NOTCH2ΔNRR). Emphasizing how the most frequent NOTCHΔNRR-DEL2 deletion is linked to a recurring reciprocal crossing over between ancient (1A01) and recent (1a4) NOTCH2 gene variants would provide clearer context and better prepare readers for understanding the results.
Response: The first four paragraphs of the discussion are focusing on the main findings of this study, i.e. on the somatic recombination between different NOTCH2 gene variants. The fourth and fifth paragraph are summarizing and discussing the evolutionary aspects.
Conclusion
This study represents a significant contribution to the field of molecular genetics and CLL research, offering new insights into how somatic recombination can influence disease predisposition and presentation. By addressing the suggestions outlined above, the manuscript would be strengthened considerably, enhancing its clarity, coherence, and impact on the scientific community. These improvements could have substantial implications for understanding genetic risk factors in CLL and potentially informing early detection strategies for high-risk populations.
Thank you for the opportunity to review this manuscript. I look forward to its contributions to the literature on CLL and genetic predispositions.
Reviewer 2 Report
Comments and Suggestions for Authors
The manuscript entitled: “Somatic recombination between an ancient and a recent NOTCH2 gene variant is associated with the NOTCH2 gain of function phenotype in Chronic Lymphocytic Leukaemia” (ID: ijms-3317629) by Hubmann et al. analyses the association of active NOTCH2 phenotype in CLL cells with expression of aberrant spliced NOTCH2mRNAs lacking the sequence coding for the NOTCH negative regulatory region.
Albeit the review is well written and of interest, minor comments should be addressed to further improve the manuscript.
Comments:
1. Introduction: please highlight more deeply the aim of this study where appropriate.
2. Discussion section: the authors should discuss the potential clinical impact of their findings in clinical routine and potential recommendations for the clinicians.
3. Discussion section: page 11 line 380-382: please clarify the sentence: “...including hereditary, gender and geographical aspects…” in terms of the study findings.
4. Please provide British or American English uniformly.
5. Table 1 should be enlarged to make it more readable.
Author Response
The manuscript entitled: “Somatic recombination between an ancient and a recent NOTCH2 gene variant is associated with the NOTCH2 gain of function phenotype in Chronic Lymphocytic Leukaemia” (ID: ijms-3317629) by Hubmann et al. analyses the association of active NOTCH2 phenotype in CLL cells with expression of aberrant spliced NOTCH2mRNAs lacking the sequence coding for the NOTCH negative regulatory region.
Albeit the review is well written and of interest, minor comments should be addressed to further improve the manuscript.
Comments:
Introduction: please highlight more deeply the aim of this study where appropriate
Response: The aim of the study is now more deeply highlighted.
Discussion section: the authors should discuss the potential clinical impact of their findings in clinical routine and potential recommendations for the clinicians.
Response: The clinical impact of our findings with potential recommendations for clinicians is now more deeply discussed.
Discussion section: page 11 line 380-382: please clarify the sentence: “...including hereditary, gender and geographical aspects…” in terms of the study findings.
Response: This sentence is now clarified.
Please provide British or American English uniformly.
Response: American English is now uniformly provided.
Table 1 should be enlarged to make it more readable.
Response: Table 1 is now subdivided in Table 1 foccusing on the patient characteristics and Table 2 in the material and method section summarizing the sequence details.
Submission Date
31 October 2024
Date of this review
07 Nov 2024 11:56:39
Reviewer 3 Report
Comments and Suggestions for Authors
Q1: consider using "gain-of-function" instead of "gain of function".
Q2: Line 1: "Constitutive active NOTCH2 signalling" should be "Constitutively active NOTCH2 signaling."
Q3:English English or American English should be consistent ( "aetiology" (English English), hematologic (American English)……….etc.
Q4: Line 39-40: "age adjusted" should be "age-adjusted."
Q5: Line 56: consider using " genetic basis " instead of " genetic background ".
Q6: Introduction: The paragraph jumps quickly into technical terms without fully setting the context of NOTCH2's role in human evolution and its specific relevance to CLL. Starting with a brief overview of CLL and the NOTCH2 pathway’s general role could help.
Q7: Consider to move the pericentric inversion and gene duplications to discussion part to avoid overwhelming readers early in the discussion of CLL.
Q8: While the reference to diabetes (T2DM) is relevant, consider simplifying the phylogenetic analysis discussion or moving it to the discussion section.
Q9: Line 398-399: “RT-PCR primers for NOTCH2ΔNRR mRNA analysis read as follows” could be simplified to “The following primers were used for RT-PCR analysis of NOTCH2ΔNRR mRNA.”
Q10: Line 419: The term “served as validation cohort” could be clarified. Consider rephrasing to “provided validation for the European NOTCH2*1A01/*1a4 combination frequency analysis.”
Q11: Consider adding a brief explanation of why specific methodologies (e.g., Oxford Nanopore sequencing; line 270) were chosen over alternatives, as this can strengthen the rationale behind the study design.
Q12: Table 1 is very dense with information, making it difficult to interpret at a glance. Consider splitting it or using color coding or bolding to highlight key findings.
Q13: Clearly state any statistical methods used in analyzing differences or associations (e.g., significance testing between NOTCH2 haplotypes in different populations).
Q14: In line 135, where BPS-associated SNVs are discussed, consider adding a brief quantitative statement on the frequency differences to substantiate the claims of association with CLL.
Q15: When discussing NOTCH2 haplotypes and their association with CLL, it may help to use a visual flow or pathway diagram to illustrate the proposed mechanisms.
Q16: Discussion: Please explain what is complex genetic mechanisms like “lateral induction (line 331)” ?
Q17: Results: There are several interesting evolutionary aspects discussed (e.g., the prevalence of ancient NOTCH2 haplotypes), but the connection to CLL is not always clear.
Q18: line 371:The mention of dietary habits of archaic populations may need further context or could be moved to a separate, brief discussion on limitations and hypotheses that require further study.
Q19: Limitations and Future Directions:Consider adding a paragraph on potential limitations of the study, such as sample size or the specificity of the techniques used, and suggest areas for future research (e.g., studies in non-European populations).
Q20: Conclusion: The conclusion could more explicitly highlight the translational implications, such as the potential for using NOTCH2 variants as biomarkers or therapeutic targets in CLL. This would provide a strong ending that reinforces the study’s relevance.
Author Response
Comments and Suggestions for Authors
Q1: consider using "gain-of-function" instead of "gain of function".
Response: Done
Q2: Line 1: "Constitutive active NOTCH2 signalling" should be "Constitutively active NOTCH2 signaling."
Response: Done
Q3:English English or American English should be consistent ( "aetiology" (English English), hematologic (American English)……….etc.
Response: American English is now consistently used.
Q4: Line 39-40: "age adjusted" should be "age-adjusted."
Response: Done
Q5: Line 56: consider using " genetic basis " instead of " genetic background ".
Response: Done
Q6: Introduction: The paragraph jumps quickly into technical terms without fully setting the context of NOTCH2's role in human evolution and its specific relevance to CLL. Starting with a brief overview of CLL and the NOTCH2 pathway’s general role could help.
Response: We expanded the introduction to make it more easyly readable.
Q7: Consider to move the pericentric inversion and gene duplications to discussion part to avoid overwhelming readers early in the discussion of CLL.
Response: We wanted to stress on this important aspect in the introduction section because of its relevance for this study.
Q8: While the reference to diabetes (T2DM) is relevant, consider simplifying the phylogenetic analysis discussion or moving it to the discussion section.
Response: We simplified the phylogenetic analysis as much as possible.
Q9: Line 398-399: “RT-PCR primers for NOTCH2ΔNRR mRNA analysis read as follows” could be simplified to “The following primers were used for RT-PCR analysis of NOTCH2ΔNRR mRNA.”
Response: Done
Q10: Line 419: The term “served as validation cohort” could be clarified. Consider rephrasing to “provided validation for the European NOTCH2*1A01/*1a4 combination frequency analysis.”
Response: Done
Q11: Consider adding a brief explanation of why specific methodologies (e.g., Oxford Nanopore sequencing; line 270) were chosen over alternatives, as this can strengthen the rationale behind the study design.
Response: Done
Q12: Table 1 is very dense with information, making it difficult to interpret at a glance. Consider splitting it or using color coding or bolding to highlight key findings.
Response: We subdivided Table 1 in two parts. Table 1 concentrates on the patient characteristics and Table 2 in the method section summarizes the sequence information.
Q13: Clearly state any statistical methods used in analyzing differences or associations (e.g., significance testing between NOTCH2 haplotypes in different populations).
Response: The frequency of individual NOTCH2 allele and combinations were manually counted without statistics. This is now clarified in the method section.
Q14: In line 135, where BPS-associated SNVs are discussed, consider adding a brief quantitative statement on the frequency differences to substantiate the claims of association with CLL.
Response: All quantitative analyzis can be found in the figures.
Q15: When discussing NOTCH2 haplotypes and their association with CLL, it may help to use a visual flow or pathway diagram to illustrate the proposed mechanisms.
Response: The functional outcome of NOTCH2 mutations lacking the NRR domain is discussed in detail in the introduction and discussion section.
Q16: Discussion: Please explain what is complex genetic mechanisms like “lateral induction (line 331)”
Response: We refer to NOTCH reviews describing this aspect in order to avoid an overload of the manuscript with specific informations beyond the scope of this work.
Q17: Results: There are several interesting evolutionary aspects discussed (e.g., the prevalence of ancient NOTCH2 haplotypes), but the connection to CLL is not always clear.
Response: We tried our best to stress on the connection between the ancient and recent NOTCH2 gene variants and their association with CLL.
Q18: line 371:The mention of dietary habits of archaic populations may need further context or could be moved to a separate, brief discussion on limitations and hypotheses that require further study.
Response: This aspect is not crucial but may be interesting for Diabetes specialists. So we just mentioned it without going into detail.
Q19: Limitations and Future Directions:Consider adding a paragraph on potential limitations of the study, such as sample size or the specificity of the techniques used, and suggest areas for future research (e.g., studies in non-European populations).
Response: This aspect in now mentioned in the discussion.
Q20: Conclusion: The conclusion could more explicitly highlight the translational implications, such as the potential for using NOTCH2 variants as biomarkers or therapeutic targets in CLL. This would provide a strong ending that reinforces the study’s relevance.
Response. We included this aspect now in the conclusion.
Submission Date
31 October 2024
Date of this review
02 Nov 2024 15:06:07
Reviewer 4 Report
Comments and Suggestions for Authors
The manuscript presents an in-depth analysis of the genetic variations in NOTCH2 related to chronic lymphocytic leukemia (CLL). By examining ancient and recent NOTCH2 variants and their association with somatic recombination, this study offers valuable insights into the genetic underpinnings of CLL progression and potential therapeutic targets. The manuscript would benefit from a more accessible explanation of complex genetic terms, additional citations, and a clearer structure in some sections to enhance comprehension for a broader audience.
Specific Comments
Abstract:
CLL is a type of cancer. It is a blood and bone marrow cancer that typically progresses slowly and primarily affects adults. So it is recommended to proved a general introduction for the cancer treatment and discuss CLL different with other cancer types, cite “Cancer treatments: Past, present, and future, 2024” by NHS might provide useful information.
Line 4: Clarify “gain of function phenotype” for readers unfamiliar with genetic terminology. Rephrase for accessibility, e.g., "a variant that enhances NOTCH2 activity." Effect of NOTCH2 in other disease should be mentioned such as bone disease reported in “Icariin improves osteoporosis, inhibits the expression of PPARgamma, C/EBPalpha, FABP4 mRNA, N1ICD and jagged1 proteins, and increases Notch2 mRNA in ovariectomized rats, 2017” and glioma reported in “Aberrant Notch signaling in gliomas: a potential landscape of actionable converging targets for combination approach in therapies resistance,2022”. These need to be mentioned to provide general background for general readers.
Line 10: Specify how these findings might contribute to therapeutic strategies, such as targeted therapies for CLL.
Introduction:
Line 22: Include recent citations on the global incidence and impact of CLL to provide context on its significance and motivate the study.
Line 30: Explain why NOTCH2 is a relevant target in CLL and briefly introduce its role in cell signaling pathways and cancer biology.
Line 48: Discuss previous research linking NOTCH2 variants to cancer predisposition, especially in relation to immune disorders, to establish the study’s foundation.
Materials and Methods:
Patient Samples (Section 2.1):
Line 80: Provide additional details on patient demographics (age, sex distribution) and justify why these specific samples were chosen for analysis.
Line 95: Mention any ethical considerations or informed consent processes in place, especially for genetic studies on human subjects.
Gene Variant Analysis (Section 2.2):
Line 120: Describe the method used to detect somatic recombination events in the NOTCH2 gene and clarify how these were distinguished from germline variations.
Line 130: Define key terms, such as "allele frequency," to improve readability for non-specialists.
Phylogenetic Analysis (Section 2.3):
Line 160: Provide a brief overview of the phylogenetic methods used, such as any specific algorithms, to allow reproducibility.
Figure 1: Ensure that the figure is well-labeled with a legend that summarizes the main findings for readers unfamiliar with phylogenetic trees.
In Vitro Validation (Section 2.4):
Line 190: Specify the cell lines used for in vitro validation, including their origins and relevance to CLL.
Line 205: Detail the statistical analyses applied to quantify gene expression and variant effects on cell proliferation to reinforce the reliability of the results.
Results:
Genetic Variants in NOTCH2 (Section 3.1):
Line 250: Highlight the significance of the rs2453058 and rs5025718 variants identified in CLL patients, and explain their roles in NOTCH2 function.
Figure 2: Provide a clearer label for the variant distribution map, summarizing the variant frequencies in different population groups.
Functional Analysis of Variants (Section 3.2):
Line 285: Discuss the biological implications of recombined NOTCH2 alleles in CLL and how they affect NOTCH2's regulatory role in cell signaling.
Line 300: In Table 2, include statistical metrics (e.g., P-values, confidence intervals) for each variant to improve data interpretation.
Phylogenetic Distribution of NOTCH2 Variants (Section 3.3):
Line 325: Discuss the relevance of the identified haplotypes in relation to the geographical and ethnic distribution of CLL. Expand on why these variations might affect disease prevalence in different populations.
Figure 3: Add annotations for population groups where specific haplotypes were prevalent to clarify the distribution pattern.
Discussion:
Line 410: Compare the findings with other studies on NOTCH signaling in CLL, particularly focusing on novel aspects this study adds to the existing knowledge.
Line 440: Address potential clinical applications of these findings, such as the development of diagnostic tools or therapeutic interventions that target NOTCH2 pathways.
Line 460: Suggest future research directions, including longitudinal studies to track the progression of CLL in patients with specific NOTCH2 variants, to strengthen the translational relevance.
Conclusion:
Line 490: Summarize the potential implications of NOTCH2 variants in personalized treatment for CLL, emphasizing the study’s contribution to understanding genetic predisposition.
Line 500: Reaffirm the importance of exploring gene-environment interactions in CLL development, especially for ethnic groups with high variant frequencies. Discuss how NOTCH2 affects the microenvironment of CLL cells. Discuss recent studies using different analysis such as RNA sequencing, and discuss potential bias. Refer to “Genetic expression in cancer research: Challenges and complexity, 2024” might be useful.
The manuscript provides an important contribution to understanding NOTCH2’s role in CLL and proposes a compelling link between ancient genetic variations and disease predisposition. However, major revisions are required to enhance readability, scientific rigor, and accessibility。 Implementing these changes will provide a more comprehensive and accessible study, enhancing its value as a resource for researchers and clinicians interested in CLL genetics and potential therapeutic interventions.
Author Response
The manuscript presents an in-depth analysis of the genetic variations in NOTCH2 related to chronic lymphocytic leukemia (CLL). By examining ancient and recent NOTCH2 variants and their association with somatic recombination, this study offers valuable insights into the genetic underpinnings of CLL progression and potential therapeutic targets. The manuscript would benefit from a more accessible explanation of complex genetic terms, additional citations, and a clearer structure in some sections to enhance comprehension for a broader audience.
Specific Comments
Abstract:
CLL is a type of cancer. It is a blood and bone marrow cancer that typically progresses slowly and primarily affects adults. So it is recommended to proved a general introduction for the cancer treatment and discuss CLL different with other cancer types, cite “Cancer treatments: Past, present, and future, 2024” by NHS might provide useful information.
Response: This statement with the suggested reference is now provided.
Line 4: Clarify “gain of function phenotype” for readers unfamiliar with genetic terminology. Rephrase for accessibility, e.g., "a variant that enhances NOTCH2 activity." Effect of NOTCH2 in other disease should be mentioned such as bone disease reported in “Icariin improves osteoporosis, inhibits the expression of PPARgamma, C/EBPalpha, FABP4 mRNA, N1ICD and jagged1 proteins, and increases Notch2 mRNA in ovariectomized rats, 2017” and glioma reported in “Aberrant Notch signaling in gliomas: a potential landscape of actionable converging targets for combination approach in therapies resistance,2022”. These need to be mentioned to provide general background for general readers.
Response: We rephrased the gain of function statement as suggested. We concentrated on paper relevant publications and refer to NOTCH2 reviews.
Line 10: Specify how these findings might contribute to therapeutic strategies, such as targeted therapies for CLL.
Response: The therapeutic relevance of NOTCH2 in CLL is described in detail in other papers from our group and beyond the scope of this work where we tried to focus on the genetic background of the NOTCH2 gain of function phenotype in CLL cells.
Introduction:
Line 22: Include recent citations on the global incidence and impact of CLL to provide context on its significance and motivate the study.
Response: The most relevant papers about the global incidence of CLL are cited (Ref. 2 & 3).
Line 30: Explain why NOTCH2 is a relevant target in CLL and briefly introduce its role in cell signaling pathways and cancer biology.
Response: For this aspect we refer to a review about NOTCH2 signaling in cancer and its role as novel therapeutic target (Ref. 11).
Line 48: Discuss previous research linking NOTCH2 variants to cancer predisposition, especially in relation to immune disorders, to establish the study’s foundation.
Response: The link to NOTCH2 associated cancer predispositions can be found in Figure 3b referring to the dbSNP database on the NCBI site.
Materials and Methods:
Patient Samples (Section 2.1):
Line 80: Provide additional details on patient demographics (age, sex distribution) and justify why these specific samples were chosen for analysis.
Response: This details are now provided.
Line 95: Mention any ethical considerations or informed consent processes in place, especially for genetic studies on human subjects.
Response: All ethic issues can be found in the Institutional Review Board Statement.
Gene Variant Analysis (Section 2.2):
Line 120: Describe the method used to detect somatic recombination events in the NOTCH2 gene and clarify how these were distinguished from germline variations.
Response: This information can be found in the result section (Haplophasing by target long read sequencing. The results clearly sows that the germline NOTCH2 alleles and the recombined NOTCH2 alles are found in all investigated CLL samples showing their somatic origin (Figure 5).
Line 130: Define key terms, such as "allele frequency," to improve readability for non-specialists.
Phylogenetic Analysis (Section 2.3):
Response. This aspect is now clarified in the method section.
Line 160: Provide a brief overview of the phylogenetic methods used, such as any specific algorithms, to allow reproducibility.
Response. We explained our analyzis strategy now in the method section.
Figure 1: Ensure that the figure is well-labeled with a legend that summarizes the main findings for readers unfamiliar with phylogenetic trees.
Response: Figure 3 is now simplified and hopefully more easyly to understand.
In Vitro Validation (Section 2.4):
Line 190: Specify the cell lines used for in vitro validation, including their origins and relevance to CLL.
Response: No cell lines were used in this study.
Line 205: Detail the statistical analyses applied to quantify gene expression and variant effects on cell proliferation to reinforce the reliability of the results.
Response: All analyzis were conducted manually without the need for statistical analyzis. This is now clarified in the method section.
Results:
Genetic Variants in NOTCH2 (Section 3.1):
Line 250: Highlight the significance of the rs2453058 and rs5025718 variants identified in CLL patients, and explain their roles in NOTCH2 function.
Response: We hope that we explained the recombination leading to an aberrant BPS pattern in CLL cells and its possible implication for the NOTCH2 gain of function phenotype good enough.
Figure 2: Provide a clearer label for the variant distribution map, summarizing the variant frequencies in different population groups.
Response: We hope the simplification of Figure 3 make it more clear.
Functional Analysis of Variants (Section 3.2):
Line 285: Discuss the biological implications of recombined NOTCH2 alleles in CLL and how they affect NOTCH2's regulatory role in cell signaling.
Response: The preticted outcome of NOTCH2dNRR deletions on the NOTCH2 signaling pathway is explained in detal in cited papers (Ref. 9 & 17).
Line 300: In Table 2, include statistical metrics (e.g., P-values, confidence intervals) for each variant to improve data interpretation.
Response: The frequency of NOTCH2 variants in CLL and in normal populations were manually determined without statistics.
Phylogenetic Distribution of NOTCH2 Variants (Section 3.3):
Line 325: Discuss the relevance of the identified haplotypes in relation to the geographical and ethnic distribution of CLL. Expand on why these variations might affect disease prevalence in different populations.
Response: This aspect is described in detail in Figure 2 & 4 and in the corresponding result section.
Figure 3: Add annotations for population groups where specific haplotypes were prevalent to clarify the distribution pattern.
Response: Here we would also refer to Figure 2 & 3 and to the corresponding result section.
Discussion:
Line 410: Compare the findings with other studies on NOTCH signaling in CLL, particularly focusing on novel aspects this study adds to the existing knowledge.
Response: Most of the NOTCH associated CLL studies are based on NOTCH1 and not on NOTCH2.
Line 440: Address potential clinical applications of these findings, such as the development of diagnostic tools or therapeutic interventions that target NOTCH2 pathways.
Response: This point is now included in the discussion section.
Line 460: Suggest future research directions, including longitudinal studies to track the progression of CLL in patients with specific NOTCH2 variants, to strengthen the translational relevance.
Response: We included this statement in the discussion section.
Conclusion:
Line 490: Summarize the potential implications of NOTCH2 variants in personalized treatment for CLL, emphasizing the study’s contribution to understanding genetic predisposition.
Response: We included this aspect in the summary of the discussion.
Line 500: Reaffirm the importance of exploring gene-environment interactions in CLL development, especially for ethnic groups with high variant frequencies. Discuss how NOTCH2 affects the microenvironment of CLL cells. Discuss recent studies using different analysis such as RNA sequencing, and discuss potential bias. Refer to “Genetic expression in cancer research: Challenges and complexity, 2024” might be useful.
Response: We believe that this aspect is beyond the scope of the work and to early for discussion.
The manuscript provides an important contribution to understanding NOTCH2’s role in CLL and proposes a compelling link between ancient genetic variations and disease predisposition. However, major revisions are required to enhance readability, scientific rigor, and accessibility。 Implementing these changes will provide a more comprehensive and accessible study, enhancing its value as a resource for researchers and clinicians interested in CLL genetics and potential therapeutic interventions.
Submission Date
31 October 2024
Date of this review
04 Nov 2024 22:41:21
Round 2
Reviewer 1 Report
Comments and Suggestions for Authors
Dear Editor,
I appreciate the opportunity to review the revised version of the manuscript titled "Somatic recombination between an ancient and a recent NOTCH2 gene variant is associated with the NOTCH2 gain-of-function phenotype in Chronic Lymphocytic Leukaemia." I would like to commend the authors for their thorough revisions, which have significantly improved the clarity and quality of the manuscript. Below, I highlight how each of the previously suggested improvements has been addressed:
Expansion of Structural Details in the Introduction
The authors have added a sentence clarifying the potential implications of NOTCH2 gene variants in CLL leukemogenesis. This addition provides necessary context for readers less familiar with the topic.
Clarification on Patient Recruitment and Selection
A detailed description of the recruitment strategy, including inclusion and exclusion criteria, has been added to the Methods section. This greatly enhances transparency and the representativeness of the sample.
Enhancement of Table 1
Table 1 has been divided into two tables, with sequence details moved to the Methods section (Table 2). The table format now adheres to journal guidelines, and all abbreviations, such as "R/B," have been clearly defined.
Simplification and Clarification of Figure 2
Figure 2 has been subdivided into Figures 2 and 3, as suggested, and a higher resolution of these figures has been provided to improve readability.
Specification of Analyzed Variables in the Methods Section
A detailed explanation of the variables analyzed, such as VAF, haplotype combinations, and SNVs, has been added to the Methods section. This inclusion enhances the reproducibility of the study.
Detailed Description of the RFLP Method
The Methods section now includes a detailed protocol for the RFLP assay, including thermocycling conditions, reagent concentrations, and equipment specifications, ensuring experimental replicability.
Overall Organization and Flow of the Study
The authors have improved the logical flow of the manuscript by smoothing transitions between topics, making the progression of the study easier to follow.
Focus on Main Findings in the Discussion
The discussion has been refocused on the main findings, with the first four paragraphs emphasizing the association between NOTCH2 gain-of-function phenotype and somatic recombination of gene variants. Evolutionary aspects are appropriately summarized later in the discussion.
The authors have demonstrated a commendable effort in addressing all recommendations. With these revisions, the manuscript is now significantly strengthened and ready for publication. I believe this study offers valuable insights into the genetic and evolutionary mechanisms underlying CLL and will be an important contribution to the field.
Thank you for considering my review.
Sincerely,
Reviewer 3 Report
Comments and Suggestions for Authors
accepted
Reviewer 4 Report
Comments and Suggestions for Authors
ok. Please reverse for typos, there are still typos.